# Prevalence and correlates of multidimensional child poverty in India during 2015–2021: A multilevel analysis

**Jalandhar Pradhan**[1]*, **Soumen Ray**[2], **Monika O. Nielsen**[2], **Himanshu**[1]

1 Department of Humanities and Social Sciences, National Institute of Technology Rourkela, Odisha, India,
2 UNICEF State Office, Bhubneshwar, Odisha, India

☯ These authors contributed equally to this work.
* jpp_pradhan@yahoo.co.uk

**Data Availability Statement:** https://www.dhsprogram.com/methodology/survey/survey-display-541.cfm.

## Abstract

Despite increasing research and programs to eradicate poverty, poverty still exists and is a far greater concern for children than adults, leading child poverty to become a political, economic, and social issue worldwide and in India. The current study aims to find variations in the prevalence of child poverty and associated factors in India during 2015–21. In the current study, we used two consecutive rounds of the National Family Health Survey (NFHS-4, 2015–16 & NFHS-5, 2019–21) to estimate child poverty (aged 0–59 months) using the Alkire-Foster method. The multilevel logistic regression analyses were performed to find the important cofounder and cluster level variation in child poverty. The results show that about 38 percent of children were multidimensionally poor in 2015–16, which reduced to 27 percent in 2019–21. The decomposition analysis suggests that contribution of nutrition domain to child poverty increases over time, whereas the standard of living substantially declines from NFHS-4 to NFHS-5. The multilevel analysis results show that the age and sex of the child, age and years of schooling of the mother, children ever born, religion, caste, wealth quintile and central, northeast, north and west regions are significantly associated with child poverty over time. Further, the variance participation coefficient statistics show that about 12 percent of the variation in the prevalence of child poverty could be attributed to differences at the community level. The prevalence of child poverty significantly declines over time, and the community-level variation is higher than the district-level in both surveys. However, the community-level variation shows increases over time. The finding suggests a need to improve the nutritional status and standard of living of most deprived households by promoting a child-centric and dimension-specific approach with more focus on PSU-level intervention should adopt in order to lessen child poverty in India.

## Introduction

Globally about 356 million children (0–17 aged) live in impoverished households, consisting of 107 million children under the age of five in 2017. Again, children are even more susceptible

**Funding:** The author(s) received no specific funding for this work.

**Competing interests:** The information of this document, however expresses author's personal observation and opinions and does not necessarily represent UNICEF's or NITR's position. This does not alter our adherence to PLOS ONE policies on sharing data and materials.

to extreme poverty, where 17.5 percent of them live in extreme poverty, compared to an estimated 7.9 percent of adults [1]. The Sustainable Development Goal (SDGs) target 1.2 suggests the international community "reduce at least half the proportion of men, women and children of all ages living in poverty in all its dimensions through national definitions by 2030" [2]. The significance of SDG target 1.2 is important, as children are first time, included in the poverty goal worldwide; focus on the multidimensional nature of poverty and poverty goal clearly to the national definition.

The global distribution of poverty is unequal, and defining poverty is a significant challenge. Although it's extensive economic success, Asia remains the world's poorest continent, with more than half of the world's impoverished people living there. Further, due to deep inequality in the south Asia region, the children are trapped in the vicious cycle of poverty and discrimination at different levels and phases, such as nutrition, health, sanitation and lag behind universal education.

The history of measuring or identifying poverty is very old for developed and developing nations [3, 4]. The traditional poverty measure was the unidimensional measure of well-being and was solely based on the minimum income or expenditure needed to maintain a subsistence level. Academic research conducted by sociologists and economists demonstrates that poverty is more than related to insufficient to feed someone or family [5]. Further, the Amartya Sen capability approach (1997) introduces the principle of social justice and well-being, a major contribution to identified poor (based on development). Sen's approach to well-being consisting two significant components 1) functioning in regards to states and actions in which individuals wish to live and 2) capacity, which refers to the possibility that the person is equipped to exercise their freedom of choice concerning different possible runs [6].

Despite these advancements, many national and international poverty measures depend on the minimum absolute measure. However, notable changes have been made to the definition and measurement of poverty in terms of the complex nature of poverty to use non-monetary or relative measures of poverty in low and middle-income countries [7–9]. There has been great discourse overuse of welfare outcome indicators presented as living standards in the context of poverty. Measurement as dwelling quality, overcrowding, access to water, sanitation, healthcare and education are utilised constantly to define poverty [10, 11]. The latest development in measuring poverty is the Multidimensional Poverty Index (MPI), constructed by the Oxford Poverty and Human Development Initiative (OPHI) in collaboration with United Nations Development Program (UNDP). The MPI covers more than 100 developing countries by using individual and household level data about health (child mortality in households and nutrition), education (school attendance and years of schooling), and standard of living (electricity, flooring, drinking water, sanitation, cooking fuel and assets) [10].

In line with the provided method for measuring poverty, child poverty measurement is still in the development process [12–22]. Child poverty is often considered the children living in income or consumption-based below-poverty-line households. However, it is widely recognised that the household-based monetary indicator cannot capture child poverty [23]. Research has concluded that the unidimensional approach to majoring in poverty can not capture the depth of child poverty as the child's need is different from adults [24]. The United Nations Convention on Child Rights (CRC) also maintains this idea for children's well-being for the betterment and adequate standard of living [25]. The multifaceted approach is necessary to measure child poverty with CRC welfare dimensions and indicators.

Children in extreme poverty are affected differently from adults, mainly by inadequate nutrition, exposure to stress, and lack of early learning, resulting in lifetime poverty. Further, the adults have direct access to many things that may help overcome the poverty state, whereas the children solely depend on their adult family members for support, care and satisfaction of

their basic needs [26]. Mounting evidence shows that healthy children are more likely to become healthy adults. Many child deprivation indexes were constructed using different domains and indicators in the developed nations, including material well-being, health, education, crime, housing, environment, family economic well-being, social relationship, economic security, exposure to risk and risky behaviour of children in need [27–29].

UNICEF developed the Multiple Overlapping Deprivation Analysis (MODA) to provide instruments for the multidimensional aspect of child poverty in terms of deprivation. MODA adopt a holistic approach to child well-being, which cannot be tackled in the sector (e.g. health, nutrition, and education), as deprivation is multifaceted and interrelated and has more adverse effects. MODA mainly focused on the child as the unit of analysis rather than the household and kept the child's life cycle approach for a different child group has another need. In MODA, there are two steps for calculating deprivation. In the first stage, the deprivation has been identified further multiple overlapping deprivations calculated to find the combination of deprivation faced by the child. However, MODA has its weight limitation due to assuming equal weight to all dimensions. In that way, a child would consider deprived in a particular dimension if he/she has been deprived of any one indicator from that dimension.

However, little known about child poverty in the context of developing countries [13–15, 20, 22]. Gordon et al. have identified eight domains of severe child deprivation, including food, safe drinking water, sanitation, health, shelter, education, information and access to services [29, 30]. A study based in Burkina Faso identified seven domains for child poverty for children aged 5–18. Countries analyse multidimensional poverty using various indicators and unit of analysis depending on how poverty is perceived [31–37].

Despite the growth in child poverty research around the globe, few studies have been conducted primarily focused on child deprivation, child poverty, or child well-being in India. Dutta (2020) utilised the MODA framework with nine dimensions, including nutrition, health, education, child protection, water, sanitation, housing, indoor air pollution and information for estimating child deprivation in India and Bangladesh, keeping the lifecycle approach [38]. In contrast, Chaurasia A. R. (2016) constructs a child deprivation index using five domains. This analysis is based on the Rapid Survey of Children (RSoC) 2013–14 for children below age 18, and the domains were survival, growth, education, protection and environment [39].

Even though child poverty research has gained attention, still child poverty is considered for children below the age of 18 [13–15, 20, 22]. However, it is well documented that every child group has their own need, and this need changes with the child's growth. Children under the age of five constitute a considerable proportion of India's population [40]. However, hardly any study explores the changes over time in child poverty and PSU-level variation in the multidimensional child poverty estimated using the standard measurement. Hence, this study aims to measure child poverty in India for 2015–16 and 2019–21 using Alkire and Foster's multidimensional approach and later explore the associated factor with cluster effect on multidimensional child poverty among children under five years.

## Data and methods

### Data

The present studies utilized two solely independent cross-sectional data from National Family Health Survey, namely NFHS-4 (2015–16) and NFHS-5 (2019–21). The survey provides essential information on crucial population and health indicators, including fertility, mortality, maternal, child and adult health, women and child nutrition, family welfare, and emerging issues like non-communicable diseases for India and each States/Union Territories. NFHS-4 survey first time provided district-level estimates for many crucial indicators, which expanded

the sample size nearly six-fold than NFHS-3, and collected information from 601509 households, 699686 women and 103525 men. NFHS-4 adopted a two-stage sampling design in rural and urban areas of India to provide district-level estimates from 28583 primary sampling units (PSU) composed of village rural areas and census enumeration blocks (CEB) in urban areas from 640 districts of India [41].

NFHS-5 survey also provided district-level (707 districts) estimates for many crucial indicators and aligned with Sustainable Development Goals (SDGs) for preparing the database for monitoring government programmes and their progress toward achieving the SDGs by 2030. NFHS-5 collected information from 636699 households, 724115 women and 101839 men. NFHS-5 adopted a two-stage sampling design in rural and urban areas of India to provide district-level estimates from 30198 primary sampling units (PSU) composed of village rural areas and census enumeration blocks (CEB) in urban areas from 707 districts of India [42]. The present research utilised data from childbirth that took place five years before the survey date. To fulfil the study's overall objective, we exclusively analyzed data of 213623 (NFHS-4, 2015–16) and 192292 (NFHS-5, 2019–21) children aged 0–59 months after eliminating pairwise missing information.

## Outcome variable

The primary outcome variable for this study is the multidimensional poverty index, as poverty can not be measured with conventional methods based on money. A three-dimension consisting of nine indicators was used to measure multidimensional child poverty and deprivation based on the SDG's goal and the availability of relevant data for the countries. The dimension are health, nutrition and living standard, and indicators are wasting underweight, immunization, child mortality in households, housing, water, sanitation, clean fuel, and information presented below Table 1 with their relevant weight.

The nutrition recommended **nutritional** assessment was constructed using WHO-Anthro to convert weight, height and child age (months) in weight-for-age (WAZ) z-score for underweight and weight-for-height Z-score (WHZ) for wasting. Children whose WAZ and WHZ were -2 standard deviations (-2 SD) from the median of the reference population were identified as underweight and wasting, respectively.

Concerning the **health** domain, two indicators were used immunization and child mortality. A child is considered to be deprived if he did not receive all basic vaccination, including

**Table 1. Dimension, indicator cutoff and weight for estimating multidimensional child poverty among children aged five years.**

| Dimension | Indicator | Deprivation Indicator with cutoff | Weight |
|---|---|---|---|
| *Nutrition* | Wasting | The child is considered deprived if his/her weight-for-height was -2 standard deviation below the reference mean | 1/6 |
| | Underweight | The child is considered deprived if his/her weight-for-age was -2 standard deviation below the reference mean | 1/6 |
| *Health* | Immunization | The child is considered deprived if he/she is not fully immunized | 1/6 |
| | Child mortality | The child is considered deprived if he/she belongs to a household with an incidence of under-5 mortality in the last five years from the survey | 1/6 |
| *Standard of living* | Water | The child is considered deprived if the household uses an unimproved water source, and the water source is located more than 30 minutes away to fetch and return to the house. | 1/15 |
| | Sanitation | The child is considered deprived if the household has no access to an improved toilet facility | 1/15 |
| | Housing | The child is considered deprived if the household material (wall, roof, and floor) is made of natural, non-permanent material. | 1/15 |
| | Cooking fuel | The child is considered deprived if exposed to indoor air pollution due to the use of solid and fossil cooking fuels inside the home. | 1/15 |
| | Information | The child is considered deprived if the mother is not exposed to specific media such as reading newspapers or magazines, listening radio and watching television weekly. | 1/15 |

(BCG, DPT, polio and measles), living in a household that reported under-five child mortality in the past five years prior to the survey. Both surveys estimated the full immunisation rate for 12–35 month children.

The **standard of living** indicators was identified as the third dimension for MPI calculation, which included five indicators: housing, drinking water, sanitation, cooking fuel, and information.

Three items measured the housing indicators: material used in the construction of the floor, wall, and roof. It is evidenced that housing condition has profound health implications on children; in some cases, it is found more than in adults. A child is considered deprived if they live in a house with dirt, sand or dug, floor or wall and roof made of natural or rudimentary materials.

The provision of clean drinking water is one of the most basic indicators for improvement in marginalized or impoverished families [43]. A child is considered deprived if the household uses an unsafe drinking water source (unprotected well, unprotected spring, and surface water of river or lake) and the water source takes more than 30 minutes to collect and return home.

Poor and unimproved sanitation plays a role in deteriorating child health, which may result in premature death in some cases [43]. Children are deprived if they have no toilet facility, share toilet, use unimproved pit latrines, or practice open defecation.

Indoor air pollution is a potential source of health risks, such as acute respiratory infections in childhood and chronic obstructive pulmonary disease. A child is considered deprived if he or she is exposed to indoor air pollution caused by solid and fossil cooking fuels inside the home.

Children and individuals need media and information to enhance their intellect and identify information sources. Therefore, it is necessary for a child should live in a household with access to mass media exposure. A child is classified as deprived in information indicators if the child lives with a mother and has not been exposed to mass media, including radio/newspaper, television and radio.

**Construction of MPI.** The multidimensional poverty indices are estimated by Alkire and Foster (AF) method. This approach provides data on various demographic accomplishments without requiring sorting or prioritization. Instead, they complement each other. The AF method offers many key decisions to the researcher, including identifying the unit of analysis, dimension, deprivation cutoff (for determining when an individual is deprived in a dimension), weights (for indicating the relative importance of various deprivation), and poverty cutoff (determining when a person is considered poor based on the amount of deprivation they experience). Due to its adaptability, the methodology can be applied in a wide variety of contexts, although its primary use has been in assessing multidimensional forms of poverty [21, 44]. The AF method utilized a dual cutoff method to recognized the poor for each indicator and aggregated them according to different dimensions. As a result, the multidimensional poverty index can be decomposed into specific dimensions and indicators, which will help support evidence-based planning by focusing on specific dimensions and indicators. The weight of the dimension is equal, and then each indicator within each dimension is equally weighted. Thus, three types of estimates are generated- the percentage of headcount poverty (H), the intensity of poverty (A) and the multidimensional poverty index ($M_0$).

The Headcount poverty measure answers the question of 'how many individuals are poor'. It is defined as H and calculated as-

$$H = \frac{q}{n}$$

Where 'q' represents the number of multidimensionally poor people, and 'n' represents the total population of the study.

Since 'H' is very sensitive to how many dimensions a poor person is deprived of, it violet a notion called 'dimensional monotonicity' given by Alkire and Foster, which holds that if a poor person becomes newly poor in an extra dimension, total poverty should rise [44]. As a result, 'H' is adjusted with the number of deprivations suffered by the poor, which reflects the intensity 'A' of poverty and calculated as:

$$A = \frac{\sum_1^q c}{q}$$

Where 'c' is the poor experienced deprivation score and the intensity of poverty is a weighted average deprivation experienced by the multidimensionally poor.

The Multidimensional poverty index is denoted by MPI and calculated as:

$$MPI = H * A$$

Thus, MPI results from the proportion of multidimensionally poor and the intensity of poverty.

**Cofounders.** The independent variable for the present study consists of socioeconomic, household, child, and mother-level factors. These factors included child age in months, child sex, female-headed household, age of mother, education level of the mother, children ever born, place of residence, religion, caste, wealth quintile (based on asset holding of household), and region (29 states and 8 UTs divided into total six regions).

## Statistical analysis

The multidimensional child poverty prevalence and association with relevant cofounders have been presented in percentages. The chi-square test with a 5% significance level has been used to show the statistical association between a categorical variable and multidimensional child poverty. A multilevel logistic regression model with a random intercept was used to understand the clustering of the respondents within the district and the PSUs or the 'community' level. Multilevel models are particularly appropriate and used for research designs where data are structured at more than one level, for example, village level, community level and state level [45]. The P-value of <0.05 is considered statistically significant. The multicollinearity test was conducted prior to multilevel analysis, and the variance inflation factor (VIF) value was found under the permissible limit of two.

**Multilevel Logistic Regression (MLR).** The multistage sampling design is characterized as the sample drawn from such a population with a hierarchical structure. Therefore, the stratified multistage sample became the norm in the sociological and demographic survey mainly due to cost, time and efficiency. For such type of sample, the data clustering should be taken into consideration during data analysis. In the present study, NFHS-4 & NFHS-5 datasets, the individual (level 1) are nested within the PSU (Level 2), which is nested within the district (Level 3). Multilevel analysis with three levels has been utilized to identify the important cofounders of child poverty at the individual, PSU and district levels (i.e. Fixed part). The multilevel logistic regression analysis allows for partitioning the variation in the outcome variable (i.e. poor child) measured at the individual level. The variance can be attributed to individual variations at the PSU and district levels [46–48].

The random intercept is used to find the random effect (clustering of individuals) at PSU and district levels. The odds ratio (OR) with 95% confidence interval (CIs) is used to present the results of the fixed effect part. The multilevel logistic regression with clustering outcome

can be presented as follows

$$\text{logit}\left(P_{ijk}\right) = \log\left(\frac{P_{ijk}}{1 - P_{ijk}}\right) = \alpha + \beta^{i} x_{ijk} + u_{jk} + v_{k}$$

Where $\log\left(\frac{P_{ijk}}{1-P_{ijk}}\right)$ is the logit function in which $P_{ijk}$ is the probability of child 'i' in the PSU 'j' and the district 'k' being poor. The '$\alpha$' is the constant, and the $u_{jk}$ and $v_k$, are the area-level residuals explained at PSU and district levels. The random effect part of the result is expressed as the variance partition coefficient (VPC) for measuring both cluster and individual-level variance [47, 49]. We employed the latent variable approach to estimate the variance partition coefficient. The current approach is helpful while analyzing binary response variables, assuming a standard logistic distribution for a binary outcome. The VPC represents the proportion of total observed individual variance in the outcome variable, attributable between cluster variables [50]. The VPC can be presented as:

$$VPC_c = \frac{\sigma_c^2 + \sigma_d^2}{\sigma_c^2 + \sigma_d^2 + \frac{\pi^2}{3}}$$

$$VPC_d = \frac{\sigma_d^2}{\sigma_c^2 + \sigma_d^2 + \frac{\pi^2}{3}}$$

Where $\sigma_c^2$ expressed as the PSU level variance and the $\sigma_d^2$ expressed as the district-level variance. The standard logistic distribution variance is $\frac{\pi^2}{3} \approx 3.29$.

**Ethics statement.** The NFHS-5 survey was conducted by the International Institute for Population Sciences (IIPS), Mumbai and received necessary ethical approval from the relevant ethics boards. We did not obtain additional ethical approval or informed consent because we accessed the anonymized NFHS-5 data available in the public domain at https://dhsprogram.com/data/available-datasets.cfm.

## Results

### Descriptive statistics and prevalence of child poverty

The socioeconomic and demographic characteristics of children 0–59 months used in the analysis are presented in Table 2. More than 60 percent of children belong to the 24–59 month age group, and 48 percent of the sample was a girl in both surveys. About 12 percent (NFHS-4) and 15 percent (NFHS-5) of children live in a female-headed household. About 31 percent of women were illiterate in NFHS-4 as compared to 22 percent in NFHS-5. The sampling distribution is quite similar in both surveys.

Overall, about 38 percent child was multidimensional poor (MDP) by using a global 33 percent cutoff in 2015–16, which declined to 27 percent in 2019–21, showing an 11 percent points decline over time. The MPI was estimated at 0.178 in NFHS-4 and 0.120 in NFHS-5 (S1 Table). Forty-five percent (NFHS-4) and 31 percent (NFHS-5) of the children were MDP in the 12–23 month age group, and the MDP is higher among male children in both surveys. In addition, the child's MDP was significantly higher in the female-headed household.

The child's MDP significantly declined with higher maternal education levels in both surveys, and illiterate mothers reported higher MDP children (57% and 45% in NFHS-4 & NFHS-5, respectively). Similarly, children whose mothers have 6-plus children reported higher MDP (60% and 44% in NFHS-4 and NFHS-5, respectively). The MDP was about two-fold in rural residing children than in urban. Similarly, scheduled tribe children reported much higher

**Table 2. Descriptive statistics (unweighted frequency) and multidimensional poverty (weighted percentage) by background characteristics among under-five children in India, 2015–2021.**

| Background characteristics | NFHS-4 | | | NFHS-5 | | |
|---|---|---|---|---|---|---|
| | N (%) | Poor* | p-value | N (%) | Poor* | p-value |
| **Age of the Child (month)** | | | | | | |
| 0–11 month | 38387 (18.0) | 32.5 | | 35051 (18.2) | 23.7 | |
| 12–23 month | 42837 (20.1) | 44.6 | | 37909 (19.7) | 30.9 | |
| 24–59 month | 132399 (62.0) | 37.2 | p = 0.000 | 119332 (62.1) | 26.5 | p = 0.000 |
| **Sex of the child** | | | | | | |
| Male | 110564 (51.8) | 38.0 | | 98937 (51.5) | 27.4 | |
| Female | 103059 (48.2) | 37.6 | p = 0.007 | 93355 (48.6) | 26.3 | p = 0.000 |
| **Female headed household** | | | | | | |
| No | 188436 (88.2) | 37.5 | | 163306 (84.9) | 26.6 | |
| Yes | 25187 (11.8) | 40.4 | p = 0.847 | 28983 (15.1) | 28.4 | p = 0.049 |
| **Mother age** | | | | | | |
| 15–24 | 67507 (31.6) | 38.2 | | 57317 (29.8) | 28.0 | |
| 24–29 | 82555 (38.7) | 36.1 | | 76834 (40.0) | 26.4 | |
| 30–34 | 41449 (19.4) | 37.6 | | 38560 (20.1) | 25.5 | |
| 35–49 | 22112 (10.4) | 44.9 | p = 0.000 | 19581 (10.2) | 28.4 | p = 0.000 |
| **Maternal education** | | | | | | |
| No schooling | 66935 (31.3) | 56.9 | | 42414 (22.1) | 44.0 | |
| 1–5 year | 30783 (14.4) | 43.4 | | 24744 (12.9) | 33.3 | |
| 6–9 years | 55432 (26.0) | 33.2 | | 52054 (27.1) | 26.4 | |
| 10+ years | 60473 (28.3) | 20.1 | p = 0.000 | 73080 (38.0) | 16.1 | p = 0.000 |
| **Children ever born/women** | | | | | | |
| 1–2 | 126839 (59.4) | 31.8 | | 122057 (63.5) | 22.6 | |
| 3–5 | 75731 (35.5) | 46.0 | | 63292 (32.9) | 33.9 | |
| 6+ | 11053 (5.2) | 60.1 | p = 0.000 | 6943 (3.6) | 43.7 | p = 0.000 |
| **Place of residence** | | | | | | |
| Urban | 50733 (23.8) | 21.5 | | 38802 (20.2) | 16.4 | |
| Rural | 162890 (76.3) | 44.2 | p = 0.000 | 153490 (79.8) | 30.6 | p = 0.000 |
| **Religion** | | | | | | |
| Hindu | 153943 (72.1) | 38.7 | | 140069 (72.8) | 27.0 | |
| Muslim | 33478 (15.7) | 36.8 | | 27311 (14.2) | 28.0 | |
| Others | 26202 (12.3) | 28.2 | p = 0.000 | 24912 (13.0) | 20.5 | p = 0.000 |
| **Caste** | | | | | | |
| Others | 46632 (21.8) | 27.9 | | 40496 (21.1) | 20.9 | |
| Scheduled caste | 40356 (18.9) | 41.2 | | 39052 (20.3) | 29.7 | |
| Scheduled Tribe | 43264 (20.3) | 56.1 | | 40371 (21.0) | 39.8 | |
| Other Backward caste | 83371 (39.0) | 37.2 | p = 0.000 | 72373 (37.6) | 25.6 | p = 0.000 |
| **Wealth quintile** | | | | | | |
| Poorest | 55801 (26.1) | 66.8 | | 52168 (27.1) | 52.4 | |
| Poor | 50464 (23.6) | 47.9 | | 44925 (23.4) | 29.9 | |
| Middle | 42836 (20.1) | 26.9 | | 37340 (19.4) | 17.6 | |
| Rich | 35690 (16.7) | 17.5 | | 32436 (16.9) | 13.5 | |
| Richest | 28832 (13.5) | 12.8 | p = 0.000 | 25423 (13.2) | 10.6 | p = 0.000 |
| **Region** | | | | | | |
| North | 40424 (18.9) | 29.4 | | 35859 (18.7) | 17.4 | |
| Central | 61365 (28.7) | 46.7 | | 48511 (25.2) | 29.8 | |

*(Continued)*

**Table 2.** (Continued)

| Background characteristics | NFHS-4 | | | NFHS-5 | | |
|---|---|---|---|---|---|---|
| | N (%) | Poor* | p-value | N (%) | Poor* | p-value |
| East | 45052 (21.1) | 46.0 | | 36977 (19.2) | 35.7 | |
| Northeast | 31749 (14.9) | 37.4 | | 30366 (15.8) | 31.2 | |
| West | 14496 (6.8) | 33.0 | | 17104 (8.9) | 26.4 | |
| South | 20537 (9.6) | 21.9 | p = 0.000 | 23475 (12.2) | 15.1 | p = 0.000 |
| **Total** | **213623 (100)** | **37.9** | | **192292 (100)** | **26.9** | |

* Multidimensional poor child (in percentage)

Sources: Author's own calculation using NFHS-4 & NFHS-5 data

MDP than any other castes in both surveys. The MDP declined with a higher wealth quintile and was lower in the richest wealth quintile (13% in NFHS-4 and 11% in NFHS-5). While making the regional comparisons, the measured MDP was found to be higher in the central region (47%), followed by the east (46%) and northeast region (37%), whereas the south region (22%) showed lower MDP children in NFHS-4. In contrast, the pattern in a slight change in NFHS-5 found the east (36%) and northeast (31%) regions children showing higher poverty, whereas the north (17%) and south (15%) regions reported lower child MDP.

## State differential in multidimensional child poverty in India

The state differential with changing prevalence of multidimensional child poverty in two rounds of the national survey, i.e. NFHS-4 and NFHS-5, is portrayed in Fig 1. Data illustrate that overall, child poverty declined significantly from NFHS-4 to NFHS-5. It shows that 27 percent of the child was MDP in NFHS-5 (2019–21), which was about 38 percent in NFHS-4 (2015–16) survey at the national level, which showed about 11 percent point decline in child poverty between these two surveys. The lowest level of child poverty was measured in the state/UTs of Puducherry (7.6%), followed by Sikkim (8%), Mizoram (8.8%), Punjab (9.7%) and Delhi (10.3%), where the highest level of child poverty was measured in the states Bihar (42.5%), followed by Jharkhand (41.3%), Assam (35.3%), Madhya Pradesh (33.1%), and Gujrat (29.8) in the lasted NFHS-5 (2019–21) survey. However, in the NFHS-4, Sikkim (8.6%) state reported the lowest child MDP, followed by Kerala (9.3%) and Chandigarh (10%), whereas Jharkhand (58%) reported a higher MDP child followed by Bihar (52.3%) and Madhya Pradesh (52.2%).

The most significant change has been observed in Rajasthan, where child poverty declined about 19 percent points from 41.7 percent to 22.4 percent in NFHS-4, followed by Madhya Pradesh (19.1 percentage points), Jharkhand (17 percentage points), Uttar Pradesh (16.1 percentage points), and Chhattisgarh (15.1 percentage points). Similarly, the lowest change has been observed in Sikkim (0.7 percentage points), followed by Goa (1.4 percentage points) and Mizoram (2.6 percentage points). However, some states/Uts like Chandigarh (2.1 percentage points), Kerala (1.9 percentage points), Lakshadweep (1.0 percentage points), Nagaland (0.7 percentage points), and Himachal Pradesh (0.3 percentage points) showed a slight increase in child poverty during NFHS-4 to NFHS-5.

## Decomposition of multidimensional child poverty in India

Multidimensional child poverty was decomposed to understand the contribution of the domain and various indicators in multidimensional child poverty in India in NFHS-4 &

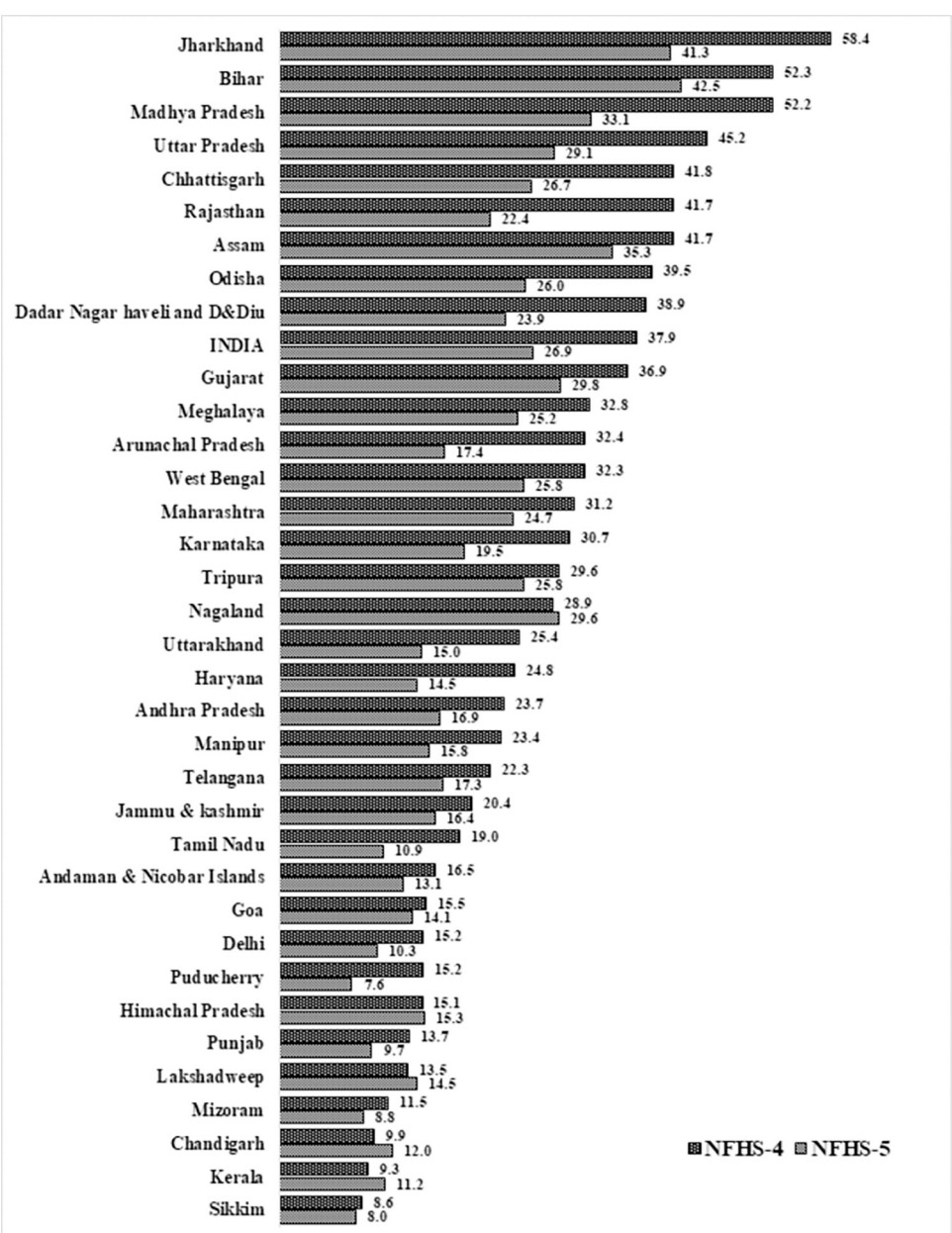

**Fig 1. State-wise pattern of the multidimensionally poor child (headcount) in the States of India, 2015–21.**

NFHS-5. Table 3 shows that among nine indicators, the underweight contributed the highest (about 30%) to multidimensional child poverty, followed by wasting (21%) in NFHS-5. A similar pattern was also observed in the NFHS-4 survey, where the highest contributor to multidimensional poverty was underweight (27.5%), followed by wasting (17%). Drinking water and child mortality in households contributed the least to multidimensional poverty at 2 percent and 2.3 percent, respectively, in NFHS-5.

The domain-wise contribution shows that the nutrition domain (44.5%) contributed the most, with a slightly lower contribution from the standard of living domain (43.6%), whereas the health domain (11.9%) contributed the least to multidimensional child poverty in NFHS-4.

**Table 3. Decomposition for contributing factor to multidimensional child poverty in India, 2015–21.**

| Domain & Indicators | NFHS-4 | NFHS-5 |
|---|---|---|
| **Nutrition** | **44.5** | **51.3** |
| Wasting | 17.1 | 21.2 |
| Underweight | 27.5 | 30.1 |
| **Health** | **11.9** | **9.7** |
| Immunization | 9.8 | 7.4 |
| Child mortality | 2.1 | 2.3 |
| **Standard of living** | **43.6** | **39.0** |
| Drinking water | 3.0 | 2.0 |
| Sanitation | 11.4 | 8.4 |
| Housing | 10.6 | 10.4 |
| Cooking fuel | 12.0 | 10.6 |
| Informatin | 6.6 | 7.6 |

Sources: Author's own calculation using NFHS-4 & NFHS-5 data

Interestingly, a similar pattern was observed in the NFHS-5 survey, where the nutritional domain contributes the highest (51%) to multidimensional poverty, followed by living standard (39%) and the least from the health domain (10%).

## Predictor of multidimensional child poverty in India

The results from multilevel logistic regression show correlate of multidimensional poverty among children 0–59 months by socioeconomic and demographic characteristics are presented in Table 4. Results show that that child age, children ever born by mother, Muslim religion, caste, and north, central and west region are more likely to have multidimensional poverty compared to their reference categories. Children from 12–23 months [AOR: NFHS-4 = 2.22; NFHS-5 = 1.71] and 24–59 month [AOR: NFHS-4 = 1.37; NFHS-5 = 1.20], were more likely to experience MDP child than 0–11 month children. The rural residence children [AOR = 1.12] are more likely to have multidimensional poverty than urban residence children in NFHS-4. Similarly, children belonging to scheduled caste [AOR: NFHS-4 = 1.15; NFHS-5 = 1.14], scheduled tribes [AOR: NFHS-4 = 1.28; NFHS-5 = 1.22] and other backward castes [AOR: NFHS-4 = 1.09; NFHS-5 = 1.08] were more likely to be MDP compared to others caste children. additionally, children belonging to central region [AOR: NFHS-4 = 1.57; NFHS-5 = 1.41], east region [AOR: NFHS-4 = 1.13; NFHS-5 = 1.41] and west region [AOR: NFHS-4 = 1.58; NFHS-5 = 1.92] were more likely to be MDP compared to the north region children.

Moreover, female children, mother age, maternal education, belonging to other religions and higher wealth quintile and south region are showing lower odds of having child poverty compared to their respective reference group. The female children [AOR: NFHS-4 = 0.93; NFHS-5 = 0.89] have showing 7% and 11% less likely to be MDP compared to male children in NFHS-4 & NFHS-5, respectively. Similarly, children from the northeast region [AOR: NFHS-4 = 0.72; NFHS-5 = 0.90] region were28% and 19% less likely to be MDP compared to the north region of India in NFHS-4 & NFHS-5, respectively.

Later the multilevel analysis finds the variation in child poverty between district and PSU levels in India. It observed that the child poverty variation is declining during NFHS-4 to NFHS-5, i.e. variance partition coefficient (VPC) at the district and PSU level, which contributes 4.4% and 10.7 percent, respectively, in NFHS-4 to the total variation in the child poverty prevalence. The variance partition coefficient (VPC) in child poverty prevalence during

**Table 4. Results of Multilevel logistic regression predicting multidimensional poverty among children under age five in India, 2015–2021.**

| | Adjusted Odds Ratio [95% CI] | Adjusted Odds Ratio [95% CI] |
| --- | --- | --- |
| **Background characteristics** | NFHS-4 | NFHS-5 |
| **Age of the Child (month)** | | |
| 0–11 month | 1.00 | 1.00 |
| 12–23 month | 2.2***[2.12: 2.27] | 1.71***[1.65: 1.78] |
| 24–59 month | 1.37***[1.33: 1.41] | 1.2***[1.16: 1.24] |
| **Sex of the child** | | |
| Male | 1.00 | 1.00 |
| Female | 0.93***[0.92: 0.95] | 0.89***[0.87: 0.91] |
| **Female-headed household** | | |
| No | 1.00 | 1.00 |
| Yes | 1 [0.97: 1.03] | 0.99 [0.95: 1.02] |
| **Mother age** | | |
| 15–24 | 1.00 | 1.00 |
| 24–29 | 0.88***[0.86: 0.91] | 0.94***[0.91: 0.97] |
| 30–34 | 0.81***[0.79: 0.84] | 0.86***[0.83: 0.89] |
| 35–49 | 0.78***[0.75: 0.82] | 0.82***[0.78: 0.86] |
| **Maternal education** | | |
| No schooling | 1.00 | 1.00 |
| 1–5 year | 0.82***[0.79: 0.84] | 0.82***[0.79: 0.85] |
| 6–9 years | 0.71***[0.69: 0.74] | 0.72***[0.68: 0.74] |
| 10+ years | 0.62***[0.60: 0.64] | 0.60***[0.58: 0.63] |
| **Children ever born/women** | | |
| 1–2 | 1.00 | 1.00 |
| 3–5 | 1.08***[1.05: 1.11] | 1.1***[1.06: 1.13] |
| 6+ | 1.24***[1.18: 1.31] | 1.22***[1.14: 1.31] |
| **Place of residence** | | |
| Urban | 1.00 | 1.00 |
| Rural | 1.12***[1.08: 1.16] | 1.01 [0.97: 1.06] |
| **Religion** | | |
| Hindu | 1.00 | 1.00 |
| Muslim | 1.06***[1.03: 1.11] | 1.17***[1.12: 1.22] |
| Others | 0.92**[0.87: 0.98] | 0.87***[0.82: 0.93] |
| **Caste** | | |
| Others | 1.00 | 1.00 |
| Scheduled caste | 1.15***[1.10: 1.19] | 1.14***[1.09: 1.19] |
| Scheduled Tribe | 1.28***[1.23: 1.34] | 1.22***[1.16: 1.28] |
| Other Backward caste | 1.09***[1.06: 1.13] | 1.08***[1.05: 1.13] |
| **Wealth quintile** | | |
| Poorest | 1.00 | 1.00 |
| Poor | 0.51***[0.50: 0.53] | 0.43***[0.41: 0.44] |
| Middle | 0.22***[0.21: 0.23] | 0.23***[0.22: 0.24] |
| Rich | 0.13***[0.12: 0.14] | 0.18***[0.16: 0.19] |
| Richest | 0.10***[0.09: 0.11] | 0.15***[0.14: 0.16] |
| **Region** | | |
| North | 1.00 | 1.00 |
| Central | 1.57***[1.41: 1.74] | 1.41***[1.29: 1.54] |

*(Continued)*

**Table 4.** (Continued)

| | Adjusted Odds Ratio [95% CI] | Adjusted Odds Ratio [95% CI] |
| --- | --- | --- |
| Background characteristics | NFHS-4 | NFHS-5 |
| East | 1.13*[1.01: 1.26] | 1.41***[1.27: 1.55] |
| Northeast | 0.72***[0.64: 0.82] | 0.90*[0.81: 0.99] |
| West | 1.58***[1.39: 1.81] | 1.92***[1.72: 2.14] |
| South | 0.93 [0.83: 1.04] | 1.05 [0.95: 1.16] |
| **Random Effect Part** | | |
| **Variance (SE)#** | | |
| District | 0.164 (0.011) [0.14: 0.19] | 0.101 (0.008) [0.09: 0.12] |
| PSU | 0.228 (0.113) [0.21: 0.25] | 0.342 (0.012) [0.32: 0.37] |
| **VPC (%)^** | | |
| Level 3 (District) | 4.4% | 2.7% |
| Level 2 (PSU) | 10.7% | 11.9% |

p*<0.05

p**<0.01

p***<0.001

#Variance expressed in standard error

^Variance Participation Coefficient

NFHS-5 declined to 2.7 percent at the district level, whereas it increased at the PSU level (11.9%).

## Discussion

Research on multidimensional poverty has been conducted globally and is also available in India [51]. However, this research mainly focused on all age groups and treated children as poor if they belonged to deprived households. Limited research has been conducted on the child poverty aspect in India. Although some workers have been carried out for children 0–17 age group using the MODA framework [38], MODA has its limitation as it depends upon the deprivation in the dimension where any child could be deprived if any indicator from a dimension deprived it considers that the child would be deprived on that dimension too.

Using the Alkire Foster method, this is the first study to estimate child poverty (under-five age group). The data of 213623 and 190916 children aged 0–59 months from the NFHS-4 (2015–16) and NFHS-5 (2019–21) survey has been utilized to obtain the prevalence and pattern of child poverty over time. The study estimates multidimensional child poverty using nine indicators from three dimensions and employs the Alkire-Foster (AF) method. Children are very vulnerable, and MPI-based research has shown that poverty among children is higher than that of adults [21]. Later the multilevel analysis was employed to obtain the district and PSU-level variance contribution to overall child poverty prevalence.

First, at the national level, about 27 percent of children were multidimensional poor in the latest NFHS-5 survey, which was 38 percent in NFHS-4. The Global report on MPI statistics for children (0–9 years) shows a higher child poverty in the neighbouring countries ranging from 28–60 percent compared to India (38% in 2015–16 and 27% in 2019–21). For instance, children from Afganistan (60% in 2015–16), Bangladesh (32.5% in 2019), Bhutan (45% in 2010), Nepal (27.9% in 2019) and Pakistan (48.6% in 2017–18) experiencing much higher multidimensional poverty. However, it also noted that the Global report on MPI calculates MPI at the household level and differentiated them by age-group whereas our analysis was performed

at individual level with relevant child indicators [52]. In addition to this, other monetary measures of poverty, such as World Bank poverty estimated based on less than $1.90 per day, failed to capture the intensity and depth of poverty among children, as the children have different needs compared to other household members. In comparison, we solely use the child-related indicator to estimate child poverty, which directly and, in some cases, indirectly affects child well-being. Result also suggests that about 42.5 percent of children from Bihar were multidimensional poor, followed by Jharkhand (41%), Assam (35%), Madhya Pradesh (33%) and Gujrat (30%) in NFHS-5. Tripathi & Yenneti [32] have also observed these states with higher multidimensional poverty.

Moreover, Puducherry, Sikkim, Mizoram, Punjab, Delhi, Tamil Nadu, Kerala, and Chandigarh had the lowest levels of multidimensional child poverty, with about 7 to 12 percent of the children being multidimensionally poor in NFHS-5. The central and northern region state like Bihar, Jharkhand, Chattisgarh, Madhya Pradesh, and Uttar Pradesh constitute a significant proportion of the Indian population [40], and have the potential to alter the multidimensional child poverty at the national level. The results indicate that many states and union territories are improving their child poverty prevalence over time. States and Union territories like Puducherry, Rajasthan, Arunachal Pradesh, Tamil Nadu, Haryana, and Uttarakhand have reduced 41–50 percent child poverty over time from 2015–16 to 2019–21. In the same tenure at the India level, child poverty was also found to decline by 11 percentage points (29%), showing improvement in many child-related indicators [41, 42]. However, some state and union territories like Chandigarh, Kerala, Lakshadweep, Nagaland and Himachal Pradesh have increased child poverty over time. Perhaps this is due to the fact that child nutrition indicators in many states and union territories have increased from NFHS-4 to NFHS-5 [41, 42]. Additionally, India's latest Global Hunger Index ranking (94 out of 107 countries) in 2020 supports this undernourished situation. Programmes such as Integrated Child Development Services Schemes (ICDS), Midday Meal Programme, and Iodine Deficiency Disorders Control Programme are targeted nutritional programmes that uplift the nutritional standard and play an important role in combating nutritional deficiencies, especially among women and children over the decade.

Second, the decomposition of multidimensional poverty by indicator suggests that among ten indicators, underweight (30%) made the most significant contributor to multidimensional poverty, followed by wasting (21%) in NFHS-5, which needs to be further addressed to improve underweight and wasting among children to reduce multidimensional child poverty in India. Two-fifths of children over the age of 0–59 months suffer from malnutrition in India, which is considered a serious problem in the face of public health [41].

The nutrition dimension contribution has increased over time (44.5% in NFHS-4 to 51% in NFHS-5), whereas the standard of living dimension decreased by 4.4 percent-points between NFHS-4 to NFHS-5 (43.6% to 39%, respectively). The results imply that the government programme improved more standard of living indicators such as improved drinking water, toilet building under Swacch Bharat Abhiyan (Clean India Movement) for improved sanitation, government aid for constructing pakka houses under different central and state government schemes, and promotion of clean cooking fuel through Ujjawala Yojana. One study finding from Ghana reveals that living standards are the most significant contributor to child poverty [53], coinciding with our study of NFHS-4, but not for NFHS-5. Different study settings, sample sizes, research designs, and survey times might explain the difference (S1 Fig).

The present study also examines the amount of variability in multidimensional child poverty using multilevel analysis for the effect of each level [48, 50]. The variation in the prevalence of multidimensional child poverty has been presented with the help of VPC. The smaller value of VPC in NFHS-4 (4.4%) demonstrates a modest variance at the district level, and a

more significant variation is observed at the community or PSU level (10.7%). This implies that the many socio-economic indicators vary from district and PSU levels. Further, the result shows that the variation in multidimensional child poverty has declined over time (4.4% in NFHS-4 to 2.7% in NFHS-5) at the district level but increased (10.7% in NFHS-4 to 11.9% in NFHS-5) at the PSU level in same duration.

The higher number of children born by mothers is positively associated with child poverty. Our finding is in line with other studies where a higher number of children in the family comprises well-being of the child and quality of care [31, 54, 55]. Perhaps this is because Indian families still favour sons, and to fulfil this, couples have many offspring to obtain their desired sex composition, even in a small family. Further, a higher number of children reduces parental attention and sometimes increases parental stress with a higher number of children. Moreover, it is advisable to consider children's age as the different stages of life associated with diverse levels of expenditure required by childcare. The Rural reside children reported higher poverty compared to urban reside children, which is in line with previous studies [29, 30, 53, 56, 57], where rural children experienced more poverty compared to urban mainly due to the information constraints in the rural area. Mother age, maternal education, and belonging to a wealthier quintile were significantly associated with child poverty. The higher mother's age, education, and wealth are major factors that improve the mother's and child's overall health. Similarly, these indicators are directly associated with the 3-A: affordability, availability and accessibility of programmes and treatment of any communicable diseases in childhood which may curb the illness and improve the child's nutritional status. The socially disadvantage population sub-group, namely the scheduled tribe children, are more likely to be multidimensionally poor compared to general caste children confirming the previous research [34, 35].

The present study has a few limitations. First, this study is based on cross-sectional information, so any causal relationship between multidimensional child poverty and its predictor could not be established. The indicator and dimension selection was challenging; however, the dimension and indicators of child poverty estimates have been selected based on prior work and research. Lastly, the present study focused on children under the age five. Moreover, some individual child samples were removed from the final analysis as the anthropometric measurements were missing (child's height/weight measurements are out of plausible limits), the pairwise removal of missing information.

## Conclusion

The study has estimated under-five child poverty in India, using Alkire and Foster's multidimensional approach. The finding of this paper reveals the significant contribution of the nutrition dimension to child poverty in India, as poverty has a negative impact on growth and educational performance at a later age and has laid a weak foundation for the future. So attempt to improve the nutritional status of children by providing healthy dietary food and an effective public distribution system (PDS) including diverse and nutritious food grains. It also stipulates the inclusion of early interventions to improve the child's nutritional status for a better future and lower child poverty. Recently Government of India launched 'POSHAN Abhiyan' in 2018 to reach the most deprived region in India with the primary aim of bringing a significant drop in country's overall national deprivation. Such policy changes by the government of India show that the government is still working hard to reach the SDG goal of ending hunger and ensuring everyone has enough food. However, such programmes have been going on for a long time with the same goal and methods, they have helped reduce the poverty indicator over time.

There are significant regional differences in experiencing multidimensional child poverty. Children in the southern regions are estimated to have lower multidimensional poverty, while those in the east, northeast and central regions of India have the highest multidimensional poverty. Therefore, more focus should be given to states and regions with higher incidence and intensity to improve the condition of overall child poverty and achieve equitable and inclusive growth in the country. Notably, being underweight, wasting, immunization, clean cooking fuel, housing condition, and sanitation are significant sources of early childhood deprivation. Finally, there were significant variations in MDP at district and PSU levels, and at PSU levels, it increased compared to district-level MDP over time. The finding of this research support an in-depth assessment to expose the causes of poverty at the district and PSU levels to improve policymaking. Additionally, the interventions may be modified in such manner that enhance their target and take into account the differences between state, district, and PSU levels.

Therefore, efforts should be made to enhance the nutritional status and standard of living of most deprived households by promoting a child-centric and dimension-specific approach and focusing on PSU-level intervention so that child poverty can be minimised and eliminated in India.

## Supporting information

**S1 Table. Headcount ratio (H), intensity (A) and multidimensional poverty index ($M_0$), India and states.**
(TIF)

**S1 Fig. Percentage of children deprived in each indicator during 2015–2021.**
(TIF)

## Acknowledgments

The authors are grateful to the Department of Humanities and Social Sciences, National Institute of Technology (NIT) Rourkela and UNICEF, Odisha, for their support and encouragement, which helped improve this research paper.

## Author Contributions

**Conceptualization:** Jalandhar Pradhan, Soumen Ray, Monika O. Nielsen, Himanshu.

**Data curation:** Himanshu.

**Formal analysis:** Himanshu.

**Investigation:** Jalandhar Pradhan.

**Methodology:** Jalandhar Pradhan, Soumen Ray, Himanshu.

**Software:** Himanshu.

**Supervision:** Jalandhar Pradhan, Monika O. Nielsen.

**Writing – original draft:** Himanshu.

**Writing – review & editing:** Jalandhar Pradhan, Soumen Ray, Monika O. Nielsen.

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
