## [Decision Letter · Decision Letter 0]

24 Oct 2022

PONE-D-22-19492Prevalence and correlates of multidimensional child poverty in India during 2015-2021: A multilevel analysisPLOS ONE

Dear Dr. Pradhan,

Thank you for submitting your manuscript to PLOS ONE. After careful consideration, we feel that it has merit but does not fully meet PLOS ONE’s publication criteria as it currently stands. Therefore, we invite you to submit a revised version of the manuscript that addresses the points raised during the review process.

Minor Revisions 

We look forward to receiving your revised manuscript.

Kind regards,

Faisal Abbas, PhD

Academic Editor

PLOS ONE

Journal Requirements:

Additional Editor Comments:

minor revision.

Reviewers' comments:

Reviewer's Responses to Questions

**Comments to the Author**

1. Is the manuscript technically sound, and do the data support the conclusions?

Reviewer #1: Yes

Reviewer #2: Yes

2. Has the statistical analysis been performed appropriately and rigorously? 

Reviewer #1: Yes

Reviewer #2: Yes

3. Have the authors made all data underlying the findings in their manuscript fully available?

Reviewer #1: Yes

Reviewer #2: Yes

4. Is the manuscript presented in an intelligible fashion and written in standard English?

Reviewer #1: Yes

Reviewer #2: Yes

5. Review Comments to the Author

Reviewer #1: The document is well written and issue is very pertinent. There is a need to research child poverty. It would be better if you add a discussion on child poverty with household poverty. As the document also mentions that the child has different needs and it may be the case that a household that we count as non-poor has poor child (multi-dimensionally).

It would be better to add the this discussion.

Secondly compare the statistics of this study with some published empirical literature so that robustness of the measurement can be highlighted.

Reviewer #2: The article is on important socioeconomic issue of child poverty titled "Prevalence and correlates of multidimensional child poverty in India during 2015-2021: A multilevel analysis". Alkire-Foster methodology is used to estimate the child poverty. The bivariate analysis is used to estimate the prevalence, and the chi-square test was carried out to show the significance level of the association between the outcome variable and its correlates. Later, multilevel logistic regression analyses were performed to find the important cofounder and cluster level variation in child poverty.

The paper is well organized. The author/s has carried out an empirical study. The subject matter of this work is of relevance to the theme of the journal, which would be helpful in the context of analysis of overall literature contribution in the field. The information presented is based on published data. Following are some specific issues/comments for improvement in the quality of the article.

1. The abstract should be in a single paragraph and clearly mentioning the policy implications emerging from it.

2. The introduction section should include significance of study on previous studies and for the policy i.e. rational of the study and it should also include literature gap. Recent studies on the topic should also be included.

3. The choice of methodology is also important. The author(s) has used standard Alkire-Foster methodology which has been used in many studies. The novelty of the methodology should be narrated and should also be mentioned that why it is specifically used.

7. Some recent literature should be added to justify the results.

8. I would like to see more discussion in the discussion section and concluding section by comparing the results with other studies.

9. The policy implications should be added more explicitly based on the results emerging from the article.

6. PLOS authors have the option to publish the peer review history of their article (what does this mean?). If published, this will include your full peer review and any attached files.

Reviewer #1: No

Reviewer #2: No

---

## [Author Response · Author response to Decision Letter 0]

4 Nov 2022

We would like to thank the reviewers for their careful and thorough reading of the manuscript and for providing insightful comments and constructive suggestions, which helped improve our manuscript's quality. All the suggestions have been addressed in the revised manuscript. The manuscript is thoroughly revised, and its final version is enclosed. Point-by-point responses to the reviewer's comments are listed below.

Note: the reviewers' and editor's comments/suggestions and their responses are incorporated in the final manuscript in tracked change mode.

Reviewer #1: 

Comment 1: The document is well written and issue is very pertinent. There is a need to research child poverty. It would be better if you add a discussion on child poverty with household poverty. As the document also mentions that the child has different needs and it may be the case that a household that we count as non-poor has poor child (multi-dimensionally). It would be better to add the this discussion.

Response: The authors are thankful for the suggestion now it has been incorporated into the revised manuscript. The changes have been done in lines 435-438.

Comment 2: Secondly compare the statistics of this study with some published empirical literature so that robustness of the measurement can be highlighted.

Response: The authors are grateful for the suggestion. As suggested, we have added findings from recent studies on child MPI(UNDP and OPHI, 2022) . The change have been done in lines 426-432.

Reviewer #2: 

The article is on important socioeconomic issue of child poverty titled "Prevalence and correlates of multidimensional child poverty in India during 2015-2021: A multilevel analysis". Alkire-Foster methodology is used to estimate the child poverty. The bivariate analysis is used to estimate the prevalence, and the chi-square test was carried out to show the significance level of the association between the outcome variable and its correlates. Later, multilevel logistic regression analyses were performed to find the important cofounder and cluster level variation in child poverty.

The paper is well organized. The author/s has carried out an empirical study. The subject matter of this work is of relevance to the theme of the journal, which would be helpful in the context of analysis of overall literature contribution in the field. The information presented is based on published data. Following are some specific issues/comments for improvement in the quality of the article.

Comment 1: The abstract should be in a single paragraph and clearly mentioning the policy implications emerging from it.

Response: The suggestion are incorporated with single paragraph abstract. Moreover, policy implications have been improved to the revised manuscript abstract section. The change have been done in lines 48-50. 

Comment 2: The introduction section should include the significance of study on previous studies and for the policy i.e. rational of the study and it should also include literature gap. Recent studies on the topic should also be included.

Response: Thank you for suggesting the literature gap with the rationale of the study has been incorporated into the revised manuscript. The changes have been done in lines 138-143. Following references have been incorporated in the revised draft. 

13. Trani JF, Biggeri M, Mauro V. The multidimensionality of child poverty: Evidence from Afghanistan. Social indicators research. 2013 Jun;112(2):391-416.

14. Singh R, Sarkar S. Children's experience of multidimensional deprivation: Relationship with household monetary poverty. The Quarterly Review of Economics and Finance. 2015 May 1;56:43-56.

15. Alkire S. Child Poverty in Bhutan: Insights from Multidimensional Child Poverty Index (C-MPI) and Qualitative Interviews with Poor Children. National Statistics Bureau,[Government of Bhutan]; 2016.

20. Alkire S, Ul Haq R, Alim A. The state of multidimensional child poverty in South Asia: a contextual and gendered view. OPHI Working Paper 127, University of Oxford; 2019.

22. Nawab T, Raza S, Shabbir MS, Yahya Khan G, Bashir S. Multidimensional poverty index across districts in Punjab, Pakistan: estimation and rationale to consolidate with SDGs. Environment, Development and Sustainability. 2022 Jan 19:1-25.

Comment 3: The choice of methodology is also important. The author(s) has used standard Alkire-Foster methodology which has been used in many studies. The novelty of the methodology should be narrated and should also be mentioned that why it is specifically used.

Response: We appreciate the suggestion. The AF method is globally utilized to measure poverty due to its adaptability and provide control to the researcher. A detailed description is added to the revised manuscript's 'Construction of MPI' section. The changes have been done in lines 217-224. Following references are cited regarding the advantages of AF method over other poverty measures.

21. Dirksen J, Alkire S. Children and Multidimensional Poverty: Four Measurement Strategies. Sustainability [Internet]. MDPI AG; 2021;13:9108. Available from: http://dx.doi.org/10.3390/su13169108

44. Alkire S, Roche JM, Ballon P, Foster J, Santos ME, Seth S. Multidimensional poverty measurement and analysis. Oxford University Press, USA; 2015.

Comment 4: Some recent literature should be added to justify the results.

Response: As suggested, now we have incrportaed few recent literature in the revised MS. 

Nawab T, Raza S, Shabbir MS, Yahya Khan G, Bashir S. Multidimensional poverty index across districts in Punjab, Pakistan: estimation and rationale to consolidate with SDGs. Environment, Development and Sustainability. 2022 Jan 19:1-25.

Alkire S. Child Poverty in Bhutan: Insights from Multidimensional Child Poverty Index (C-MPI) and Qualitative Interviews with Poor Children. National Statistics Bureau,[Government of Bhutan]; 2016.

UNDP and OPHI. Global Multidimensional Poverty Index 2022: Unpacking deprivation bundles to reduce multidimensional poverty. Available from :https://hdr.undp.org/system/files/documents/hdp-document/2022mpireportenpdf.pdf

Comment 5: I would like to see more discussion in the discussion section and concluding section by comparing the results with other studies.

Response: The discussion section has been updated with more relevant research work. The changes have been done in lines 524-529, 538-541.

Comment 6: The policy implications should be added more explicitly based on the results emerging from the article.

Response: Thank you for the suggestion. The policy implication has been extended in line with the study findings.

---

## [Decision Letter · Decision Letter 1]

4 Dec 2022

Prevalence and correlates of multidimensional child poverty in India during 2015-2021: A multilevel analysis

PONE-D-22-19492R1

Dear Dr. Pradhan,

We’re pleased to inform you that your manuscript has been judged scientifically suitable for publication and will be formally accepted for publication once it meets all outstanding technical requirements.

Kind regards,

Faisal Abbas, PhD

Academic Editor

PLOS ONE

Additional Editor Comments (optional):

accept with minor revision.

Reviewers' comments:

Reviewer's Responses to Questions

**Comments to the Author**

1. If the authors have adequately addressed your comments raised in a previous round of review and you feel that this manuscript is now acceptable for publication, you may indicate that here to bypass the “Comments to the Author” section, enter your conflict of interest statement in the “Confidential to Editor” section, and submit your "Accept" recommendation.

Reviewer #1: (No Response)

Reviewer #2: All comments have been addressed

2. Is the manuscript technically sound, and do the data support the conclusions?

Reviewer #1: Yes

Reviewer #2: Yes

3. Has the statistical analysis been performed appropriately and rigorously? 

Reviewer #1: Yes

Reviewer #2: Yes

4. Have the authors made all data underlying the findings in their manuscript fully available?

Reviewer #1: Yes

Reviewer #2: Yes

5. Is the manuscript presented in an intelligible fashion and written in standard English?

Reviewer #1: Yes

Reviewer #2: Yes

6. Review Comments to the Author

Reviewer #1: 1. The manuscript need to be carefully proof read for spellings and grammar correction such as Line 416.

2. The random variation at PSU level and district needs further clarification based on the case of India. Why does the PSU level variation is lesser than district level variation. Identify district level variables that are responsible for random variation and may be incorported in the model for future research.

3. Why did not the researcher test the region level variation (uban/rural)?

4. Diagnostic tests are missing.

Reviewer #2: (No Response)

7. PLOS authors have the option to publish the peer review history of their article (what does this mean?). If published, this will include your full peer review and any attached files.

Reviewer #1: No

Reviewer #2: No

---

## [Editor Report · Acceptance letter]

13 Dec 2022

PONE-D-22-19492R1 

Prevalence and correlates of multidimensional child poverty in India during 2015-2021: A multilevel analysis 

Dear Dr. Pradhan:

I'm pleased to inform you that your manuscript has been deemed suitable for publication in PLOS ONE. Congratulations! Your manuscript is now with our production department. 

Kind regards, 

on behalf of

Dr. Faisal Abbas 

Academic Editor

PLOS ONE